# Associations between Menopausal Hormone Therapy and Colorectal, Lung, or Melanoma Cancer Recurrence and Mortality: A Narrative Review

**DOI:** 10.3390/jcm12165263

**Published:** 2023-08-12

**Authors:** Gabriel Fiol, Iñaki Lete, Laura Nieto, Ana Santaballa, María Jesús Pla, Laura Baquedano, Joaquín Calaf, Pluvio Coronado, Esther de la Viuda, Plácido Llaneza, Borja Otero, Sonia Sánchez-Méndez, Isabel Ramírez, Nicolas Mendoza

**Affiliations:** 1Spanish Menopause Society (Asociación Española para el Estudio de la Menopausia—AEEM), 28036 Madrid, Spain; luisignacio.letelasa@osakidetza.eus (I.L.); doctora1983@gmail.com (L.N.); lbaquedanome@hotmail.com (L.B.); jcalaf@santpau.es (J.C.); pcoronadom@gmail.com (P.C.); esterdelaviuda@gmail.com (E.d.l.V.); pllanezac@hotmail.com (P.L.); borja.otero@gmail.com (B.O.); ssanmen@gmail.com (S.S.-M.); isaramirezp@telefonica.net (I.R.); nicomendoza@telefonica.net (N.M.); 2Spanish Society of Medical Oncology (Sociedad Española de Oncología Médica), 28001 Madrid, Spain; anasantaballa@gmail.com; 3Gynecological Oncology Section of the Spanish Society of Gynecology and Obstetrics (Sección de Oncología Ginecológica de la Sociedad Española de Ginecología y Obstetricia), 28036 Madrid, Spain; mjpla@bellvitgehospital.cat

**Keywords:** cancer survival, menopausal hormone treatment

## Abstract

*Objective:* to develop eligibility criteria for use in non-gynecological cancer patients. *Methods:* We searched all the articles published in peer-reviewed journals up to March 2021. We utilized the PICOS standards and the following selection criteria: menopausal women with a history of non-gynecological and non-breast cancer who underwent hormone replacement therapy (HRT) using various preparations (oestrogens alone or in combination with a progestogen, tibolone, or tissue selective oestrogen complex) and different routes of administration (including oral, transdermal, vaginal, or intra-nasal). We focused on randomized controlled trials as well as relevant extension studies or follow-up reports, specifically examining recurrence and mortality outcomes. *Results:* Women colorectal cancer survivors who use MHT have a lower risk of death from any cause than those survivors who do not use MHT. Women who are skin melanoma survivors using MHT have a longer survival rate than non-MHT survivors. There is no evidence that women lung cancer survivors who use MHT have a different survival rate than those who do not use MHT. *Conclusions:* MHT is safe for women who have a history of colorectal, lung, or skin melanoma cancers.

## 1. Introduction

The information gathered over the past two decades regarding the impact of menopausal hormone therapy (MHT) holds promise for a more reliable and successful long-term approach to managing menopause symptoms and potential complications. These complications may encompass osteoporotic fractures, cognitive impairment, cardiovascular conditions, and an overall enhanced quality of life [1,2,3,4]. With these findings in mind, international organizations have determined that the benefits of MHT outweigh the associated risks for healthy postmenopausal women experiencing symptoms, particularly if the therapy commences within a decade of menopause or before the age of 60 [5,6].

Nevertheless, there is currently a lack of guidelines that offer recommendations on prescribing MHT to postmenopausal women who have medical conditions that could affect its use. In the realm of contraception, a comprehensive resource known as the “WHO Medical Eligibility Criteria” provides guidance on various medical conditions, aiding the scientific community in ensuring the safe use of contraceptive methods [7].

Among the factors that significantly influence the suitability of any form of MHT is a history of non-gynecological cancers such as colorectal, lung, or melanoma. The concern lies in the potential recurrence of these diseases, thus impacting the decision to administer MHT. Moreover, a considerable number of women who have survived these cancers have undergone treatments that intensify menopausal symptoms due to the effects of cancer therapies. This further underscores the importance of providing effective remedies for these individuals, with MHT being one of the noteworthy options [8].

In previous articles, we have addressed the study of MHT in gynecological cancer, but we considered studying the repercussions of MHT in other cancers that, due to their high prevalence and especially their possible relationship with estrogens, could be of interest.

According to the Globocan 2022 study, colon cancer was the second most frequent cancer in women, with an incidence of 16.2 per 100,000 women [9].

The WHI study showed a significantly lower incidence of colon cancer after MHT (HR 0.6, CI 0.43–092), noting that the mechanism by which this reduction occurs is not clear [10]. We currently know that estrogen action is mediated by two types of receptors, ERα and Erβ with a similar interaction, but ERα interacts with c-Jun and c-Fos of the activator protein-1 (AP-1) complex and the specificity of the factor transcription protein 1 (SP1) more frequently than ERβ [11]. It is specified that ERβ has a suppressive effect on tumor progression in patients with some cancers and has shown therapeutic potential in the management of these malignancies [12]. In the normal colon, ERβ plays a fundamental role in maintaining physiology and the immune response [13]. Stimulation of these receptors with estrogen, even with flavonoids [14], has the capacity to stimulate the tumor suppressor function in colon cancer, favoring this protective effect [15].

The third most common cancer in women is lung cancer, with an incidence of 14.6 per 100,000 women [9]. The relationship between lung cancer and tobacco use is well established, although in some studies endogenous estrogen stimulation or hormonal treatments have also been associated with the development of this neoplasia, perhaps associated with a greater susceptibility of women to tobacco carcinogenesis [16], mainly due to the induction of the CYP1B1 enzyme, thus interfering with estrogen metabolism and causing greater formation of reactive oxygen species with an oncogenic nature [17]. The Women’s Health Initiative trial concluded that estrogen and progesterone treatment in postmenopausal women did not increase the incidence of lung cancer [10]. However, other investigators have shown in vitro that estrogen induces the spread of lung cancer cells from non-small cells and tumor growth [18], while anti-estrogen treatment strategies can decrease tumor size, cell growth, and proliferation [19].

Although gender differences have been shown in the incidence of melanoma, there are doubts about the estrogenicity of this divergence.

At the molecular level, it has been described how, unlike the breast, which has a predominant presence of ERα, which promotes DNA transcription, there is a greater presence of ERβ in the skin, which inhibits this expression and significantly reduces tumor capacity [20].

Epidemiologically, hormonal treatment in women with melanoma has been studied with initially divergent results. While previous studies do not contraindicate hormonal contraception in this tumor [21,22], other authors show a greater risk of estrogen treatment in postmenopausal women that is not found if combined estrogen-progestogen therapy is administered [23].

Based on these biologic and clinical reasons, we have chosen colon, lung, and melanoma cancer as the objects of our review.

The objective of this report is to develop eligibility criteria for the use of this method in non-gynecological (colorectal, lung, or melanoma) cancer patients, similar to those established for contraceptive methods. A panel of experts from various Spanish scientific societies (the Spanish Menopause Society, the Sociedad Española de Ginecología y Obstetricia, and the Sociedad Española de Oncología Médica) met in order to draw up a series of evidence-based recommendations.

## 2. Methods

The study was registered at www.prospero.org (accessed on 28 April 2020) (registration number CRD42020166658) and is part of the “Eligibility criteria for MHT project” (Appendix A).

### 2.1. Inclusion Criteria

The systematic review included randomized controlled trials and related extension studies or follow-up reports. Additionally, we included observational studies (with special interest in population-based cohorts or large case-control studies), including a control group of non-users. Studies considered eligible were those including menopausal women of any age receiving MHT in melanoma, colorectal, and lung cancer survivors.

We considered studies that evaluate any MHT preparation (oestrogens alone or combined with a progestogen, tibolone, or tissue selective oestrogen complex) with any route of administration (oral, transdermal, vaginal, or intra-nasal). The impact of MHT was compared with placebo or non-treatment controls.

### 2.2. Selection of Studies

We conducted a comprehensive literature search using various databases, including MEDLINE (via PubMed), The Cochrane Library (CENTRAL), and EMBASE (via embase.com), covering articles from their inception until the most recent date. A tailored search strategy, comprising controlled vocabulary and specific search terms for each cancer type, was designed for each database. When necessary, validated filters were applied to retrieve relevant study designs.

To ensure the unbiased inclusion of studies, two independent researchers screened the retrieved references, and any disagreements were resolved by a member of the expert panel. The panel members were actively involved in evaluating the suitability of the included studies and suggested additional relevant articles that might have been overlooked.

The review’s scope was guided by the PICOS (Population, Intervention, Comparators, Outcomes, Study Design) criteria, along with specific procedures for study selection and literature synthesis. The inclusion criteria focused on menopausal women of any age with non-gynecological and non-breast cancer receiving MHT, encompassing various MHT preparations and administration routes. The selected outcomes of interest were recurrence and mortality, and only randomized controlled trials, extension studies, or follow-up reports were considered.

### 2.3. Data Extraction, Synthesis, and Risk of Bias Assessment

Data extraction, synthesis, and risk of bias assessment were performed by the panel, consisting of the study’s authors, following methodological guidelines from the Cochrane collaboration [24]. The reporting adhered to the Preferred Reporting Items for Systematic Reviews and Meta-Analyses (PRISMA) statement [25]. For observational studies, the ROBINS I tool was adapted to evaluate confounding variables, selection bias, outcome measures, and attrition [26].

The evidence was synthesized following PRISMA guidelines with a narrative approach focusing on the outcomes of interest to explore the association between MHT and cancer survivors. When appropriate, pooled analyses were conducted using the Mantel–Haenszel method and the random effects model within the RevMan software statistical package (v 5.3.5) [27]. To assess the certainty of the evidence for each outcome of interest, explicit judgments were made based on GRADE criteria [28], classifying quality as high (A), moderate (B), low (C), or very low (D), considering factors like risk of bias, inaccuracy, inconsistency, lack of directionality, and publication bias. In cases where no direct evidence was available but plausibility or clinical experience with indirect evidence existed, the panel reached consensus decisions labeled “Expert opinion.”

### 2.4. Evidence for the Decision Framework and Eligibility Criteria

To formulate recommendations transparently and logically, an evidence-to-decision (EtD) framework was employed [29], which aided the panel in considering relevant aspects necessary for decision-making. The framework involved integrating findings from the reviews, including the magnitude of MHT effects on recurrence and mortality, certainty ratings of the evidence, and data from other sources.

As per the WHO international nomenclature, the proposed MHT eligibility criteria have been unified into four categories:Category 1: no restrictions on MHT use;Category 2: benefits outweigh the risks;Category 3: risks generally outweigh the benefits;Category 4: MHT should not be used.

## 3. Results

### 3.1. Colorectal Cancer

Five observational studies on the use of MHT in women with colorectal cancer (CRC) were identified: two prospective [30,31] and three retrospective [32,33,34] cohorts totaling 5510 women receiving some form of MHT (Table 1).

For this report, a combined evaluation was made of the results of four of these studies [30,31,32,33], which showed a significant reduction in the proportion of women who died of any cause among those who received MHT compared with those who were untreated (RR 0.75, 95% CI: 0.65–0.86; Figure 1a). The results were not statistically significant for the proportion of women with a cause of death attributable to CRC (RR 0.72, 95% CI: 0.52–1.02; Figure 1b) because of the discrepancy between estimates from prospective (RR 0.75, 95% CI: 0.46–1.23) and retrospective studies (RR 0.63, 95% CI: 0.40–0.97).

### 3.2. Lung Cancer

Five studies were included on the effect of MHT on lung cancer: three retrospective [35,36,37,38] and one prospective [39] collected in a meta-analysis [40] with a total of 1054 women with lung cancer who used MHT compared with 1528 who did not (Table 2).

No publication bias was observed among these studies. The sensitivity analysis result showed that the overall results were stable. The meta-analysis revealed that patients who received hormone replacement therapy survived 5 years longer than patients not treated (RR = 0.346; 95% CI: 0.216–0.476; *p* < 0.001).

### 3.3. Cutaneous Melanoma

A prospective [41] cohort study was identified that analyzed the effect of MHT on 206 women with cutaneous melanoma (Table 3). After 10 years of follow-up, adjusted analysis of the results showed increased survival among women who received MHT after melanoma surgery (HR 0.17, 95% CI: 0.04–0.62). An estimate of the data provided by the study showed a statistically significant reduction in the risk of death in those who received MHT (RR 0.06, 95% CI: 0.01–0.41; Figure 2).

Regarding recurrences, this study showed no difference in the proportion of women with MHT who suffered recurrences (RR 0.74, 95% CI: 0.14 to 3.95; Figure 2). The quality of evidence for this outcome is very low due to the serious methodological limitations of the study and the inaccuracy of the estimate attributable to a small sample size.

## 4. Discussion

On the basis of a combined analysis of the researched literature, it is concluded that MHT is safe, in terms of recurrence and/or mortality, in patients who have suffered some of the most frequent non-gynecological cancers (colorectal, lung, or melanoma).

### 4.1. Why Is This Report Important?

There is considerable confusion regarding the appropriateness of MHT in women with cancer, primarily because of the fear of recurrence or increased mortality that may occur with its use.

Women who have suffered from cancer often present earlier and more intense menopausal symptoms due to the effects of some of their treatments, which seriously affect their quality of life. On many occasions, the long-term risks overlap with those suffered by women with premature ovarian insufficiency, thus extending the suitability of MHT [8].

### 4.2. Strengths

This is the first published work to present a systematic review to analyze the recurrence and mortality of MHT in women survivors of the most common non-gynecological cancers (colorectal, lung, and melanoma).

It is also the first time that categories of evidence (eligibility criteria) have been distinguished for the use of MHT in these patients using the strictest methodological tools.

### 4.3. Limitations

The quality of the evidence is low overall. Many studies include the generic use of MHT without distinguishing between dose, formulation, or route of administration.

### 4.4. Clinical Evidence

#### 4.4.1. Colorectal Cancer

The recommendation is based on analyzing five observational studies: two prospective cohorts [30,31] and three retrospective studies [32,34]. There are no randomized studies, although the population studied is large, mainly due to the high prevalence of this tumor.

The results point to an increased survival rate, although differences in the means do not reach statistical significance.

This risk reduction is evident during treatment but not after the end of treatment. Treatments with estrogen alone appear to be associated with better outcomes than those with estrogen and progestin.

There are many limitations, including a lack of randomized studies, case collection, BMI, knowledge of the socio-economic profile, and treatment variability.

Results regarding disease-free survival are not sufficiently consistent to draw conclusions.

#### 4.4.2. Lung Cancer

The recommendation is based on analyzing five observational studies (three retrospective studies [35,36,37,38] and one prospective study [39]) pooled in a meta-analysis [40].

Although most of the results point to reduced survival in female MHT users, particularly smokers, other retrospective studies show better survival in MHT users, especially female smokers.

The limitations include the absence of randomized studies, different dosages, a low prevalence of the disease, and a low number of cases, along with a lack of knowledge of the socio-economic profile.

There are no available data on recurrence rates.

#### 4.4.3. Cutaneous Melanoma

The recommendation is based on a single cohort study [41] with a small sample of women (*n* = 206) and a long follow-up (10 years on average).

There is evidence of longer overall survival in this single study, with no differences in disease-free survival. In addition, no differences were found in the various histological types or the depth of the lesion. However, there were differences in tumor ulceration.

The main limitations are the absence of randomized studies, only one follow-up study, and the small sample size. There are no studies to either support or counter these data. A better evolution of the disease in younger women remains to be clarified.

### 4.5. Cancer Risk in Healthy MHT Users

#### 4.5.1. Colorectal Cancer

The results of the WHI study show that the use of equine conjugated estrogens both alone and in combination with medroxyprogesterone was associated with a lower risk of colon cancer [10]. Further studies have confirmed these results with reductions in the risk of cancer (RR 0.81, *p* = 0.005) and the risk of death from this cancer (RR 0.63, *p* = 0.002) [42]. An 18 year follow-up of women in the WHI study has shown no difference in mortality between treated and untreated women [43].

#### 4.5.2. Lung Cancer

Most of the studies aimed at assessing the risk of lung cancer in MHT users have reported positive results in reducing the risk of this type of cancer. Age, smoking, comorbidities, and family history have been considered as risk factors in these comparisons, reporting a significant reduction among users compared with non-users (RR 0.80; 95% CI: 0.70–0.93; *p* = 0.009), although no significant differences in mortality rates were found [44]. Another recent meta-analysis of 13 studies has also shown a reduced risk in users of MHT (RR 0.95, 95% CI: 0.91–0.99, I = 30.8%, *p* = 0.137) [45]. Additionally, while co-factor variability is a major limitation in this assessment, the evidence suggests a protective effect of MHT on the incidence of lung cancer. Recent studies suggest very heterogeneous and complex mechanisms of action in the carcinogenesis and development of lung cancer as far as gender, sexual hormones, and nicotine are concerned. Some data confirm the likelihood of a protective effect of estrogens once the tumor has been triggered by the carcinogenic effect of smoking [46].

#### 4.5.3. Melanoma

After reviewing the literature, the use of MHT has been associated with a significantly increased risk of melanoma (RR, 1.28; 95% CI: 1.17–1.41) [47], for both estrogens alone (RR, 1.37; 95% CI: 1.22–1.52), and estrogen-progestogen combinations (RR, 1.23; 95% CI: 1.13–1.34; *p* = 0.15) after five years of use [48]. These same authors reported significant increases in the risk of melanoma for both oral and vaginal estrogens, while no such relationship was found with estroprogestagen combination therapy (RR 0.91; 95% CI: 0.70–1.19) [23]_._ No differences were found when assessing estrogen doses, sun exposure, or age.

### 4.6. Future Research

Our report has, however, identified some important areas for improvement in future research. We expect that the results will contribute to the development of studies that further examine the safety and efficacy of MHT for treating menopausal symptoms in non-gynecological and breast cancer survivors. Larger RCTs should be conducted over a longer follow-up period to evaluate the various MHT strategies.

## 5. Conclusions

Female colorectal cancer survivors who use MHT have a lower risk of mortality from any cause than survivors who are non-users (Category 1C).

Female cutaneous melanoma survivors who use MHT have a longer survival than non-users (Category 2C).

There is no evidence that female lung cancer survivors who use MHT have a different likelihood of survival compared to non-users (Category 2C).

Although MHT use has been associated with a significantly increased risk of melanoma, it has also been shown that female survivors of cutaneous melanoma who use MHT have a prolonged survival rate compared to non-users. On this basis, we consider women with a history of melanoma eligible for MHT. (Category 2C).

## Figures and Tables

**Figure 1 jcm-12-05263-f001:**
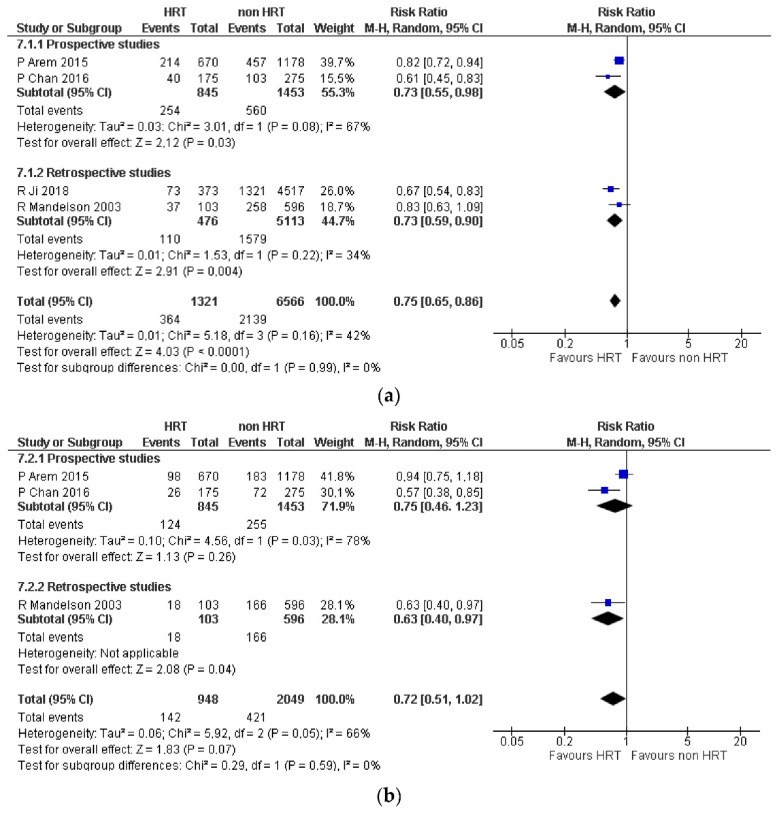
(**a**) Colorectal cancer. Mortality from any cause (data from prospective and retrospective studies). Summary of pooled risk estimates. The association between HRT use in women with colorectal cancer and all-cause mortality. (**b**) Colorectal cancer. Mortality attributable to colorectal cancer (data from prospective and retrospective studies). Summary of pooled risk estimates. The association between HRT use in women with colorectal cancer and colorectal cancer mortality. Note. HR: hazard ratio; CI: confidence intervals; HRT: hormone replacement therapy [30,31,32,33].

**Figure 2 jcm-12-05263-f002:**
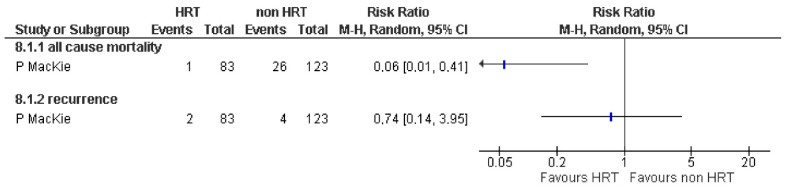
Melanoma. Mortality (from any cause) and recurrence. Summary of pooled risk estimates. The association between HRT use in women with melanoma and all cancer mortality. Note. HR: hazard ratio; CI: confidence intervals; HRT: hormone replacement therapy [41].

**Table 1 jcm-12-05263-t001:** Study characteristics of included studies. Colorectal cancer.

Study	Study Period	Country	Age	Number of Participants	Stage	Grade	MHT Type	MHT Recency	CRC Death MHT User vs. Non User	All-Cause Mean Death MHT User vs. Non User	Mean Follow-Up Year
Prospective cohort											
Chan et al. (2006) [30]	1976–2004	USA	62.2–65.7	834	22.3% stage I, 26.1% stage II, 25.5% stage III, 15.6% stage IV, 10.5% unknown	22.3% stage I, 26.1% stage II, 25.5% stage III, 15.6% stage IV, 10.5% unknown	E, E + P	Current former	Current 0.64 (0.47–0.88) former 1.05 (0.79–1.40)	Current 0.74 (0.56–0.97) former1.00 (0.78–1.30)	5–10
Arem et al. (2015) [31]	1995–2001	USA	50–71	2053	30.5% localized, 31.3% regional or distant, 38.2% unknown	12.0% well-differentiated, 57.5%; moderately-differentiated, 0.9%; undifferentiated, 29.6% unknown	E, E + P	Current former	Current 0.76 (0.59, 0.97) former 1.03 (0.72, 1.47)	Current 0.79 (0.66, 0.94) former1.13 (0.89, 1.43)	7.7
Retrospective cohort											
Mandelson et al. (2003) [32]	1980–1998	USA	50–79	699	NR	NR	E, E + P	Current	0.59 (0.35–0.97)	0.77 (0.54–1.09)	5.33
Ji et al. (2018) [33]	2006–2015	Sweden	45–69	5626	23.7% stage I, 27.8% stage II, 36.2% stage III, 12.3% stage IV	NR	E, E + P	Current	0.74 (0.62–0.88); *p* = 0.0006	0.70 (0.60–0.82); *p* < 0.0001	5.4
Slattery et al. (1999) [34]	1991–1998	USA	50–79	801	35.4% local, 53.2% regional, 11.4% distant	NR	E, E + P	Current to stop less than 5 years	504 (62.9) 297 (37.1) 0.6 (0.4 ± 0.9)	0.7 (0.5 ± 0.9)	4

**Table 2 jcm-12-05263-t002:** Study characteristics of included studies. Lung cancer.

Study	Study Period	Country	Age	Number of Participants	Stage	Smokers	MHT Type	Median Overall Survival	Median Overall Survival with MHT Smokers/Non-Smokers
Retrospective cohort									
Ganti et al. (2006) [35]	1994–1999	USA	31–93	498	I: 26% II: 21% IIIA: 11% IIIB: 8% IV: 28%	86%	All types	Never used MHT 79 months; 95% CI: 65–95 months MHT 39 months; 95% CI: 35–77 months *p* < 0.02	Smoker and MHT smoker; and used/smoke and no MHT 39 vs. 73 months; *p* < 0.03). Non-smoker and MHT/Non-smoker and not MHT; 92 vs. 98 NS
Ayeni et al. (2009) [36]									
Katkoff et al. (2014) [37]	2001–2005	USA	17–74	485	Local: 33.6% Regional: 33.4% Distant: 33.0	Current or former 92.3%	Estrogen only: 99 Estrogen plus progesterone: 85	Median survival time, MHT 80.0 m No MHT 37.5 m *p* < 0.001	Never smoked and MHT vs. no MHT: 17 (7.4)/ 20 (7.9) Current smokers and MHT vs. no MHT: 126 (54.8)/165 (65.2) NS
Huang et al. (2009) [38]	1995–2005	USA	37–90	648	I: 20.8% II: 4.8% III: 30.1% IV: 37.4% Unknown stage: 6.9%	61.9%	All types	MHT/no MHT 16.4 vs. 10.5 NS	Smoker and MHT/Non-smoker and MHT 11.3 vs. 16.9 months *p* < 0.03 Smoker and MHT/Non-smoker and MHT NS
Prospective cohort									
Clague et al. (2014) [39]	1995–1996	USA	NR	727	Localized: 153 Regional: 51 Lymph nodes: 73 Regional and lymph nodes: 33 Distant: 365 Unknown: 52	543 (74.69%)	Estrogen: 188 Estrogen + progesterone: 176	MHT: 21.4 m No MHT: 15.6 m *p* = 0.002	Used MHT vs. never used MHT (HR) Never smoked: 1.23 (0.58–2.63) Former smokers: 0.74 (0.50–1.10) Current smokers: 0.44 (0.26–0.75)
Meta-analysis									
Li et al. (2020) [40]				No HRT: 1054 HRT: 1528				With MHT, survival increased time for 5 years (ES = 0.346; 95% CI 0.216–0.476; *p* < 0.001)	

**Table 3 jcm-12-05263-t003:** Study characteristics of included studies on melanoma.

Study	Study Period	Country	Age	Number of Participants	Type	HRT Type	CRC Death/HRT User	All-Cause Mean Death/MHT User	Mean Follow-Up Year
Prospective cohort									
MacKie et al. (2004) [41]	1990–1995	Scotland	46–59	206	Ulceration: Yes, 5 (6.2); 21 (17.8) 0.017 patients with tumors 1 mm thick: 42 (50.6); 58 (47.2); 0.627 Patients with superficially spreading melanoma: 60 (73.2); 84 (69.4); 0.846 Nodular/polypoid melanoma: 15 (18.3); 25 (20.7); Lentigo maligna melanoma: 4 (4.9); 4 (4.3); Acral/mucosal melanoma: 1 (1.2); 2 (1.7); Other and unspecified melanomas: 2 (2.4); 6 (5.0).	21 oestrogen 62 oestrogen/progesrterone	MHT: 1 No MHT: 22	MHT: 0 No MHT: 4	19

## Data Availability

Data are available from https://aeem.es/criterios-de-elegibilidad-de-la-thm/ (accessed on 28 April 2020).

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
