# Peer review of "Associations between Menopausal Hormone Therapy and Colorectal, Lung, or Melanoma Cancer Recurrence and Mortality: A Narrative Review"

_jcm, 2023, doi:10.3390/jcm12165263_

Round 1

Reviewer 1 Report

This topic is very important in gynecologic area .  MHT is major a problem in women treated for cancer, because of  it is not clear this therapy whether is increased recurrence or mortality rates of related cancer.

-  in line 7  HMT , is MHT ?

- in lines 32-33, these compilications............. It is not understandable and

it includes both complications and benefit 

- in line 44-45, sentence  should be corrected, also  no dot at the end of sentence

-in line 129 , according PICOS criteria,  there is no 'comparators' in your article. It should be non-MHT group

- in figure Ia  retrospective studies are shown  as prospective studies in table 1,  so there is discordance. 

- Data in table one is very complex. It is not clear MHT receiving  and non-MHT groups and there is no data regarding cancer recurrences . So in the title, word of  'recurrence'  is not necessary. 

-in line 200-201 sentence is too long , so it is not understandable 

- in line 243, 3 prospective 2 retrospective studies should  be according to table one.

- in line 249  this data is not clear according to your study results. İt should be re-evaluated.

- The title of this article should be changed, This study has not met systematic reviews and meta-analysis. Because there is no enough data. This study may be narrative review. 

it is necessary  some revision . 

Author Response

Rewiever 1

This topic is very important in gynecologic area.  MHT is major a problem in women treated for cancer, because of it is not clear this therapy whether is increased recurrence or mortality rates of related cancer.

-  in line 7 HMT , is MHT ? It has been changed

- in lines 32-33, these complications............. It is not understandable and it includes both complications and benefit  

We have changed this sentence: “These complications may encompass osteoporotic fractures, cognitive impairment, cardiovascular conditions, and overall enhanced quality of life"

- in line 44-45, sentence should be corrected, also no dot at the end of sentence  

We have changed this sentence

-in line 129, according PICOS criteria, there is no 'comparators' in your article. It should be non-MHT group 

The included articles compare the evolution of the studied cancers (colon, lung and melanoma) after MHT versus patients with these neoplasm without MHT. No comparisons with other treatments are in the focus of this paper.

- in figure Ia retrospective studies are shown as prospective studies in table 1,  so there is discordance. 

It has been changed

- Data in table one is very complex. It is not clear MHT receiving and non-MHT groups and there is no data regarding cancer recurrences. So in the title, word of 'recurrence' is not necessary. 

we have included the term “recurrence” because it is in our objectives to know this parameter, although we recognize that perhaps we could have indicated the term “evolution”

-in line 200-201 sentence is too long, so it is not understandable. Valorar cambiar por la siguiente frase " The meta-analysis revealed that patients that received hormone replacement therapy had an increased survival to 5 years in comparison with patients not treated (RR=0.346; 95% CI: 0.216 202 to 0.476; p<0.001)" 

We have changed this sentence.

- in line 243, 3 prospective 2 retrospective studies should  be according to table one. 

It has been changed

- in line 249  this data is not clear according to your study results. İt should be re-evaluated. Cambiaría la frase por: " Better outcomes appear to be associated with estrogens alone than with estrogen with progestin treatments".

We have changed this sentence to clarify it: “Treatments with estrogens alone appear to be associated with better outcomes than those with estrogen and progestin”.

- The title of this article should be changed, This study has not met systematic reviews and meta-analysis. Because there is no enough data. This study may be narrative review.  

Although the literature is scarce on this subject, we have carried out an exhaustive review with strict methodology and Cochrane support, but we recognize that your comment is very appropriate

Thank you very much for your comments that have been of great interest to clarify the results of this review.

Reviewer 2 Report

Study is about MHT (formely known as HRT) safety in non-gynaecological cancer survivers patients (lung, colorectal and melanoma). 
We do know that MHT slightly increase the risk of gynaecological cancers however risks and benefits evaluation should be individualised and creating a set of eligibility criteria is a great contribution to the field. 

You mentioned in abstract section that you aim to develop a set of eligibility criteria but you did not mention anything about this in your conclusion section. It would be helpful if you clearly mention and add a paragraph about this. 

In discussion section you mentioned that MHT in safe in those categories, but in the beggining of you manuscript you mentioned - “ Women who have survived colo-rectal cancer under hormone replacement therapy (HRT) experience a decreased risk of mortality from any cause compared to those who do not use HRT”

From this paragraph we can understand the not only hormone therapy is safe but also increase the dissease-free interval and should be counted as a treatment.
Please clarify this aspect and also use either MHT or HRT to prevent confusions.

You discussed about smoking and lung cancer. Please also add a discussion about MHT and smoking and if this can reduce or completely cancel the efficacy of orally administered estrogens.

If MHT has been asaociated with an increased risk of melanoma, this should be clearly stated in conclusion section  

Author Response

Study is about MHT (formely known as HRT) safety in non-gynaecological cancer survivers patients (lung, colorectal and melanoma). 
We do know that MHT slightly increase the risk of gynaecological cancers however risks and benefits evaluation should be individualised and creating a set of eligibility criteria is a great contribution to the field. 

You mentioned in abstract section that you aim to develop a set of eligibility criteria but you did not mention anything about this in your conclusion section. It would be helpful if you clearly mention and add a paragraph about this.  

The aim of this report was to develop eligibility criteria for the use of MHT in patients with non-gynecologic cancer (colorectal, lung, or melanoma), similar to those established for contraceptive methods. A panel of experts from various Spanish scientific societies (Spanish Society of Menopause, Spanish Society of Gynecology and Obstetrics and Spanish Society of Medical Oncology) met to develop a series of evidence-based recommendations. These eligibility criteria are endorsed by different national and international scientific societies1 and published to be available to doctors planning to prescribe MHT to women2.

  1. Ramirez I, de la Viuda E, Calaf J, Baquedano L, Coronado P, Llaneza P, Nieto V, Otero B, Sánchez S, Mendoza N. Universidad de Granada. Criterios de Elegibilidad de la Terapia Hormonal de la Menopausia.
  2. Mendoza N, Ramírez I, de la Viuda E, Coronado P, Baquedano L, Llaneza P, Nieto V, Otero B, Sánchez-Méndez S, de Frutos VÁ, Andraca L, Barriga P, Benítez Z, Bombas T, Cancelo MJ, Cano A, Branco CC, Correa M, Doval JL, Fasero M, Fiol G, Garello NC, Genazzani AR, Gómez AI, Gómez MÁ, González S, Goulis DG, Guinot M, Hernández LR, Herrero S, Iglesias E, Jurado AR, Lete I, Lubián D, Martínez M, Nieto A, Nieto L, Palacios S, Pedreira M, Pérez-Campos E, Plá MJ, Presa J, Quereda F, Ribes M, Romero P, Roca B, Sánchez-Capilla A, Sánchez-Borrego R, Santaballa A, Santamaría A, Simoncini T, Tinahones F, Calaf J. Eligibility criteria for Menopausal Hormone Therapy (MHT): a position statement from a consortium of scientific societies for the use of MHT in women with medical conditions. MHT Eligibility Criteria Group. Maturitas. 2022 Aug 30;166:65-85. doi: 10.1016/j.maturitas.2022.08.008. Epub ahead of print. PMID: 36081216

In discussion section you mentioned that MHT in safe in those categories, but in the beggining of you manuscript you mentioned - “ Women who have survived colo-rectal cancer under hormone replacement therapy (HRT) experience a decreased risk of mortality from any cause compared to those who do not use HRT”From this paragraph we can understand the not only hormone therapy is safe but also increase the dissease-free interval and should be counted as a treatment. 

To experience a decreased risk of mortality from any cause in users of MHT does not necessarily mean that this advantage was obtained by having treated colorectal cancer. There are many other benefits linked to MHT that could contribute to survival.

There is indeed evidence of a positive effect of estrogens on colorectal cancer, but it has never been tested as an adjuvant to the conventional approach.

Please clarify this aspect and also use either MHT or HRT to prevent confusions. 

It has been changed

You discussed about smoking and lung cancer. Please also add a discussion about MHT and smoking and if this can reduce or completely cancel the efficacy of orally administered estrogens. 

Our study addresses the effect of MHT on lung cancer survivors.

As it is logical to expect, these patients have, in the majority, quit smoking.

Thus no studies are stratifying between smoking and non-smoking.

Moreover, the relationship between gender, sexual hormones and lung cancer is very complex. It can vary at different stages of the disease and reproductive life, as recently reviewed. by Hammouz et al1.

But, since your comment suggests a need for clarification on this topic, we have added a new paragraph in the "discussion" section on lung cancer

“Recent studies suggest a very heterogeneous and complex mechanism of action in the carcinogenesis and development of lung cancer as gender, sexual hormones and nicotine are concerned. Some data confirm the likelihood of a protective effect of estrogens once the carcinogenic effect of smoking has triggered the tumour”

1.- Hammouz RY, Orzechowska M, Anusewicz D, Bednarek AK. X or Y Cancer: An Extensive Analysis of Sex Differences in Lung Adenocarcinoma. Curr Oncol. 2023;30(2):1395-1415. Published 2023 Jan 18. doi:10.3390/curroncol30020107

" If MHT has been asaociated with an increased risk of melanoma, this should be clearly stated in conclusion section. 

We have introduced a clarification in this regard:

" Although MHT use has been associated with a significantly increased risk of melanoma, it has also been shown that female survivors of cutaneous melanoma who use MHT have more prolonged survival than non-users. On this basis, we consider women with a history of melanoma as eligible for MHT. Category 2C"

Thank you very much for your comments that have been of great interest to clarify the results of this review.

Round 2

Reviewer 1 Report

Your manuscript should be narrative review

(At the title , systematic review and meta-analysis should be narrative review.)

Author Response

Rewiever 1

This topic is very important in gynecologic area.  MHT is major a problem in women treated for cancer, because of it is not clear this therapy whether is increased recurrence or mortality rates of related cancer.

-  in line 7 HMT , is MHT ? It has been changed

- in lines 32-33, these complications............. It is not understandable and it includes both complications and benefit  

We have changed this sentence: “These complications may encompass osteoporotic fractures, cognitive impairment, cardiovascular conditions, and overall enhanced quality of life2

- in line 44-45, sentence should be corrected, also no dot at the end of sentence  

We have changed this sentence

-in line 129, according PICOS criteria, there is no 'comparators' in your article. It should be non-MHT group 

The included articles compare the evolution of the studied cancers (colon, lung and melanoma) after MHT versus patients with these neoplasm without MHT. No comparisons with other treatments are in the focus of this paper.

- in figure Ia retrospective studies are shown as prospective studies in table 1,  so there is discordance. 

It has been changed

- Data in table one is very complex. It is not clear MHT receiving and non-MHT groups and there is no data regarding cancer recurrences. So in the title, word of 'recurrence' is not necessary. 

we have included the term “recurrence” because it is in our objectives to know this parameter, although we recognize that perhaps we could have indicated the term “evolution”

-in line 200-201 sentence is too long, so it is not understandable. Valorar cambiar por la siguiente frase " The meta-analysis revealed that patients that received hormone replacement therapy had an increased survival to 5 years in comparison with patients not treated (RR=0.346; 95% CI: 0.216 202 to 0.476; p<0.001)" 

We have changed this sentence.

- in line 243, 3 prospective 2 retrospective studies should  be according to table one. 

It has been changed

- in line 249  this data is not clear according to your study results. İt should be re-evaluated.

We have changed this sentence to clarify it: “Treatments with estrogens alone appear to be associated with better outcomes than those with estrogen and progestin”.

- The title of this article should be changed, This study has not met systematic reviews and meta-analysis. Because there is no enough data. This study may be narrative review.  

Although the literature is scarce on this subject, we have carried out an exhaustive review with strict methodology and Cochrane support, but we recognize that your comment is very appropriate

Thank you very much for your comments that have been of great interest to clarify the results of this review.
